# Efficacy of Hip Strengthening on Pain Intensity, Disability, and Strength in Musculoskeletal Conditions of the Trunk and Lower Limbs: A Systematic Review with Meta-Analysis and Grade Recommendations

**DOI:** 10.3390/diagnostics12122910

**Published:** 2022-11-23

**Authors:** Angélica de F. Silva, Laísa B. Maia, Vanessa A. Mendonça, Jousielle M. dos Santos, Ana C. Coelho-Oliveira, Joyce N. V. Santos, Leticia L. V. Moreira, Rodrigo de O. Mascarenhas, Gabriele T. Gonçalves, Vinícius C. Oliveira, Leonardo A. C. Teixeira, Amandine Rapin, Ana C. R. Lacerda, Redha Taiar

**Affiliations:** 1Postgraduate Program in Rehabilitation and Functional Performance (PPGReab), Universidade Federal dos Vales do Jequitinhonha e Mucuri (UFVJM), Diamantina 39100-000, Brazil; 2Department of Physiotherapy, Universidade Federal dos Vales do Jequitinhonha e Mucuri (UFVJM), Diamantina 39100-000, Brazil; 3Postgraduate Program in Health Sciences (PPGCS), Universidade Federal dos Vales do Jequitinhonha e Mucuri (UFVJM), Diamantina 39100-000, Brazil; 4Faculté de Médecine, Université de Reims Champagne Ardennes, UR 3797 VieFra, 51097 Reims, France; 5MATIM, Université de Reims Champagne Ardenne, 51100 Reims, France

**Keywords:** hip strengthening, rehabilitation, musculoskeletal conditions, pain intensity, disability, hip strength

## Abstract

To investigate the efficacy of hip strengthening on pain, disability, and hip abductor strength in musculoskeletal conditions of the trunk and lower limbs, we searched eight databases for randomized controlled trials up to 8 March 2022 with no date or language restrictions. Random-effect models estimated mean differences (MDs) with 95% confidence intervals (CIs), and the quality of evidence was assessed using the GRADE approach. Very low quality evidence suggested short-term effects (≤3 months) of hip strengthening on pain intensity (MD of 4.1, 95% CI: 2.1 to 6.2; two trials, n = 48 participants) and on hip strength (MD = 3.9 N, 95% CI: 2.8 to 5.1; two trials, n = 48 participants) in patellofemoral pain when compared with no intervention. Uncertain evidence suggested that hip strengthening enhances the short-term effect of the other active interventions on pain intensity and disability in low back pain (MD = −0.6 points, 95% CI: 0.1 to 1.2; five trials, n = 349 participants; MD = 6.2 points, 95% CI: 2.6 to 9.8; six trials, n = 389 participants, respectively). Scarce evidence does not provide reliable evidence of the efficacy of hip strengthening in musculoskeletal conditions of the trunk and lower limbs.

## 1. Introduction

The stability of the lumbopelvic complex is often claimed to depend on gluteal muscle efficiency in the sagittal, transverse, and frontal planes [1,2,3]. Weakness of the gluteal muscles was previously described to be responsible for changes in the kinetics and kinematics of the trunk, hips, and/or lower limbs, leading to instability [4,5]. Moreover, their weakness and inadequate neuromuscular function have been reported to be a potential risk factor [5] for the occurrence of musculoskeletal health conditions in the trunk and lower limbs (e.g., patellofemoral and low back pain) associated with patients’ pain intensity, disability, and reduced hip strength [6,7]. In this context, the strengthening of the gluteus muscles (hip strengthening) is often recommended in clinical practice to improve function in musculoskeletal conditions of the trunk and lower limbs [2].

Preliminary basic and clinical research have suggested effects of exercise focusing on hip strengthening [2,6,7]. However, its efficacy in improving pain intensity, disability, and/or hip strength in different musculoskeletal conditions of the trunk and lower limbs and the certainty of the evidence is still unclear because of limited evidence [8,9,10]. For instance, previous systematic reviews, including nonrandomized controlled trials and/or case studies, suggested that hip strengthening enhances the effects of other conventional therapies on pain intensity and disability in chronic conditions (i.e., patellofemoral and low back pain) [9,11,12]. Therefore, this systematic review of randomized controlled trials aimed to investigate the efficacy and quality of the evidence of hip strengthening on pain intensity, disability, and gluteus strength in people with musculoskeletal health conditions of the trunk and lower limbs compared with placebo, sham, waiting list, or no intervention (control). Furthermore, we investigated whether hip strengthening combined with other active interventions enhances the effects of the other active interventions.

## 2. Materials and Methods

### 2.1. Search Strategy and Inclusion Criteria

This systematic review of randomized controlled trials followed the Cochrane Handbook for Systematic Reviews of Interventions [13]. Furthermore, this systematic review is reported based on the Preferred Reporting Items for Systematic Reviews and Meta-analyses (PRISMA) checklist (Appendix A) [14]. The protocol was registered at PROSPERO (CRD42021227725) and at the Open Science Framework (DOI 10.17605/OSF.IO/3MCW8).

The search strategy was conducted in the MEDLINE, COCHRANE (Central Register of Controlled Trials and Database of Systematic Reviews), AMED, EMBASE, CINAHL, SPORTDISCUS, and PEDRO databases up to 8 March 2022 without date or language restrictions. The detailed search strategy is presented in Appendix A in the Addenda, and the descriptors were related to ‘randomized controlled trial’ and ‘hip exercise’. In addition, we hand-searched previous systematic reviews identified in the field for potentially relevant full texts.

To be included, randomized controlled trials had to investigate the efficacy of any hip strength exercise training, defined according to the American College of Sports Medicine (ACSM) [14]. The population of interest was patients with any nontraumatic musculoskeletal health conditions of the trunk and/or lower limbs. Trials investigating pathological conditions (e.g., neurological disorders, rheumatoid arthritis, and tumors) were excluded. The comparator of interest (control) was placebo, sham, waiting list, or no intervention to investigate the isolated effects of the intervention. In addition, we included trials investigating whether hip strengthening combined with other active interventions enhances the effects of the other active interventions. The outcomes of interest were neuromuscular function (e.g., muscle strength, power, and muscle activation of the gluteus muscle), pain intensity (evaluated by any numerical perception rating scale (NPRS) or visual analog scale (VAS)), and disability (evaluated by any valid instrument) [15,16,17].

### 2.2. Trial Selection and Methodological Quality Assessment

After the search strategy, the identified references were exported to an Endnote^®^ file, and duplicates were removed. Then, two independent reviewers (AFS and JMS) screened the titles and abstracts and assessed potential full texts considering our eligibility criteria. The eligible full texts were included in the systematic review. A third reviewer (LBM) resolved between-reviewer discrepancies.

The methodological quality of the included trials was assessed independently by two reviewers (AFS and JMS) using the 0–10 Physiotherapy Evidence Database (PEDRO) scale [18,19]. The PEDRO scale is widely used to evaluate the risk of bias of trials in physiotherapy [18] and is valid and reliable [20]. When possible, we used scores that were already available at the PEDRO database (https://www.pedro.org.au/ accessed on 31 March 2020) [18,19].

### 2.3. Data Extraction

Two independent reviewers (AFS and JMS) extracted characteristics and outcome data from the included trials. A third reviewer (LBM) resolved between-reviewer discrepancies. The extracted characteristics included: participants (i.e., gender, age, and setting); a description of the intervention of interest (i.e., types and dosages) and comparators; outcomes; instrument measures; and follow-ups. Outcome data extracted at short, medium, and long terms included sample sizes, means, and standard deviations (SDs) for all groups of interest. Postintervention scores were preferably used [4,21]. The short-term effect was considered a follow-up up to three months after the baseline, the medium-term effect was considered a follow-up over three months and less than twelve months after the baseline, and the long-term effect was considered a follow-up of at least 12 months after the baseline. When more than one time point was available at the short, medium, or long terms, the point closer to the end of the intervention was considered [22,23]. All outcome data were imputed from means and/or SDs following the Cochrane recommendations [24,25]. Moreover, when the same outcome was measured with different scales, we converted them to a similar scale before pooling (i.e., an 11-point pain intensity scale and a 101-point disability scale) [24].

### 2.4. Data Analysis

When possible, the meta-analysis was conducted using a random-effects model (DerSimonian and Laird method), and the weighted mean difference (MD) with a 95% confidence interval (CI) is presented for each specific health condition in forest plots. Adjusted means were compared by Z test, and the effects were considered significant if the *p*-value ≤ 0.05. We interpreted the clinical relevance by comparing the estimated effect sizes and 95% CIs with the minimal clinically important differences (MCID) [25] of ≥10% for the pain intensity and disability scales [3]. Data from individual trials were reported when pooling was not possible.

Two independent reviewers (AFS and JMS) assessed the quality of the evidence using the GRADE methodology [26,27], and a third reviewer (LBM) solved the between-reviewer discrepancies. The four-level quality of the evidence ranged from high to very low quality, with very low quality evidence meaning that the estimate is very uncertain [26,27]. The quality of the evidence in the review began with high-quality evidence, and it was downgraded by one level based on four domains: (1) risk of bias (downgraded if greater than 25% of the analyzed participants were from trials with a high risk of bias, which we defined as a median PEDRO score < 6 out of 10) [28]; (2) inconsistency of results (downgraded if significant heterogeneity was present on visual inspection or the I2 value was >50%); (3) imprecision (downgraded if <400 participants were analyzed; <200 was considered very serious imprecision and the quality was downgraded by two levels) [29]; and (4) publication bias (we planned to evaluate publication bias using the visual inspection of funnel plots and the Egger’s test, adopting α = 0.1; however, this was not possible because of the small number of included trials; i.e., <10 trials were analyzed) [23,25].

We planned sensitivity analyses to investigate sources of heterogeneity such as poor methodological quality (trials scored < 6 on the 0−10 PEDRO scale) [25]. All analyses were conducted using the Comprehensive Metanalysis software, version 2.2.04 (Biostat, Englewood, NJ, USA).

## 3. Results

A total of 2028 titles and abstracts were screened, and 172 potential full texts were evaluated for our eligibility criteria. Nine original trials (n = 503 participants) met the inclusion criteria and were included in the review. The main reasons for excluding potential full texts were because they were not randomized controlled trials (n = 10), they did not study the condition of interest (n = 123), and they did not report the comparison of interest (n = 30) (Figure 1). Detailed reports of the excluded texts are presented in Appendix A.

### 3.1. Study Characteristics

The nine original trials were published between 2010 and 2021: two from Europe; four from South America; two from Asia; and one multicenter trial. All trials investigated nontraumatic musculoskeletal health conditions: patellofemoral pain [30,31,32] and low back pain [33,34,35,36,37,38]. The sample sizes of the included trials ranged from 20 to 90 participants and consisted of participants aged between 22 and 61 years old. Five trials included only females [30,31,32,34,35], two trials included both sexes [33,38], and two trials did not present sex distributions [36,37]. Three trials investigated the efficacy of hip strengthening compared with control (no intervention), and six trials investigated whether hip strengthening combined with other active interventions enhanced the effects of the other active interventions. The outcomes included NPRS [16,17,18,30,33,38], VAS [31,32,35,36,37], and measured pain intensity [32]. Self-reported disability was assessed by the Oswestry Disability Index (ODI) [33,34,35,36,38], the Lower Extremity Functional Scale (LEFS), the Anterior Knee Pain Scale (AKPS) [30], the Roland–Morris questionnaire [37], and the Western Ontario and McMaster Universities osteoarthritis index (WOMAC) [31]. Hip strength was assessed by a handheld isometric dynamometer and a force dynamometer [32,35,37]. Only short-term effects (i.e., ≤3 months) were investigated. Hip-strengthening programs differed in the duration of the program (ranging from 2 to 8 weeks), the duration of the session (ranging from 20 to 50 min per session), and the frequency per week (ranging from 2 to 3 times per week). Two trials [35,38] did not report the frequency or the duration of the sessions. The detailed characteristics of the nine trials are presented in Table 1.

### 3.2. Methodological Quality of the Included Trials

The PEDRO scores of the included trials ranged from 4 to 9 points, with a median of 6 points out of 10. Four of the nine trials scored at least 6 points on the 0–10 PEDRO scale. The main issues were a lack of participant blinding (n = 9; 100%), a lack of therapist blinding (n = 3; 33.3%), a lack of assessor blinding (n = 3; 33.3%), and the absence of intention-to-treat analysis (n = 4, 44.4%). The detailed methodological quality of the included trials is presented in Table 2.

### 3.3. Summary of Evidence for the Short-Term Effects of Hip Strengthening on Pain Intensity, Disability, and Hip Strength in People with Musculoskeletal Health Conditions of the Trunk and Lower Limbs

The nine included trials investigating short-term effects (i.e., ≤3 months) provided very low quality evidence for pain intensity, disability, and hip strength. The main reasons for downgrading the quality of the evidence were imprecision (three times), inconsistency (three times), and risk of bias (three times). It was not possible to conduct planned sensitivity analyses because of the small number of included trials.

### 3.4. Pain Intensity

For pain intensity (11-point pain scale), we found very low quality evidence suggesting a clinically relevant effect of hip strengthening on patellofemoral pain when compared with control groups (MD = 4.1 points, 95% CI: 2.1 to 6.2; two trials, n = 48 participants). Very low quality evidence also suggested that hip strengthening did not enhance the effects of other active interventions for patellofemoral pain (MD = 0.4 points, 95% CI: 0.9 to 1.7; one trial, n = 41 participants). For low back pain, we found very low quality evidence suggesting a clinically relevant effect of hip strengthening plus other active therapy (MD = −0.6 points, 95% CI: 0.1 to 1.2; five trials, n = 349 participants) (Figure 2).

### 3.5. Disability

For disability (101-point disability scale), we found very low quality evidence suggesting no effect of hip strengthening on patellofemoral pain when compared with control groups (MD = 29.3 points, 95% CI: 9.2 to 67.8; two trials, n = 48 participants). Very low quality evidence also suggested that hip strengthening may enhance the clinically relevant effects of other active interventions on disability in low back pain (MD = 6.2 points, 95% CI: 2,6 to 9,8; six trials, n = 389 participants) but not in patellofemoral pain (95% CI: 10.7 to 10.7; one trial, n = 41 participants; Figure 3).

### 3.6. Hip Strength

Very low quality evidence suggested an effect of hip strengthening on hip strength compared with control in patellofemoral pain (MD = 3.9 N, 95% CI: 2.8 to 5.1; two trials, n = 48 participants) and low back pain (MD = −1.1 points, 95% CI: 4.7 to 6.8; one trial, n = 70 participants) (Figure 4).

## 4. Discussion

This systematic review investigated the effectiveness of hip strengthening on pain intensity, disability, and hip strength in musculoskeletal conditions of the trunk and lower limbs (low back pain and patellofemoral pain). We found very low quality evidence supporting hip strengthening in musculoskeletal conditions for the short term. The results suggested a promising short-term isolated effect of hip strengthening on pain intensity for patellofemoral pain and an additional effect of the intervention on pain intensity and disability when combined with another active interventions for low back pain. Although uncertain, the evidence also suggested a short-term isolated effect of hip strengthening on hip strength in patellofemoral pain and low back pain. Our findings imply the need for high-quality randomized controlled trials to improve the certainty of estimates, particularly in the long term.

Despite extensive research to address all possible interventions aimed at strengthening the hip in different musculoskeletal health conditions and different populations, the literature presents a lack of evidence. For all investigated outcomes (i.e., pain intensity, disability, and hip strength) the included trials were classified as having a low methodological quality and a very low quality for the level of evidence due to imprecision, risk of bias, and inconsistency when being carefully evaluated by GRADE [27,29]. Of note, we sought the best methodology to adopt and strictly followed the protocol, with high precision and reliability in the search, extraction, and interpretation of data by reviewers, and updated the literature with more recent studies.

The hip strength exercise modalities adopted here, classified into resistance or strength, strictly followed the ACSM criteria [14]. Of note, a previous review [39] included trials with different hip exercise modalities (e.g., stretching) not specifically for strength gain and used another active intervention as a comparator. Thus, our data analysis and inclusion criteria may have led to the lower between-study heterogeneity observed in our estimates compared with the previous reviews. Furthermore, previous systematic reviews included nonrandomized controlled trials [12,40] and diverged from the scope of the study population [19,39].

Previous reviews [39,40] only assessed the influence of hip strengthening on improving pain intensity and disability and did not assess postintervention hip strength gain, although the literature makes it clear that muscle weakness and inadequate neuromuscular function are the main risk factors for the occurrence of musculoskeletal health conditions in the trunk and lower limbs [5]. Notably, even in a very short-term strength training program, we expected increases in hip strength because of neuromuscular changes. Thus, despite the studies showing a positive short-term effect, the methodologies used to assess hip strength did not use a gold standard device, such as an isokinetic dynamometer (i.e., the most recommended device compared to the portable isometric dynamometer) [41]. In addition, the studies failed to report on load prescription, including the type, duration, frequency, intensity, and progression of load, and only two studies mentioned the use of a maximum repetition test for load prescription [11].

The minimal clinically important change in pain intensity using the visual analogue scale for musculoskeletal conditions of the lower limbs has been reported to be at least 1.3 points [42]. When analyzing the trial results, on average the isolated short-term effect of hip strengthening for patellofemoral pain was effective in reducing pain by around 4.1 points for the evaluated studies. Similarly, the addition of hip-strengthening exercises to another intervention on disability for low back pain resulted, on average, in a 15 point decrease in the Oswestry disability index. Of note, the minimal clinically important change using the Oswestry disability index for disability in low back pain has been reported to be at least 12.88 points [43]. Only two of the nine included studies [31,32] assessed isolated hip strengthening. Our results showed that the short-term isolated effect of hip strengthening for patellofemoral pain was effective in increasing hip strength by around 4.1%. In this regard, the minimal clinically important change for abductor muscle strength obtained by handheld isometric dynamometer has been reported to be at least 5.3% [44]. In brief, our analysis showed that the short-term effects were also clinically important, in addition to statistically significant, for the outcomes and instruments that were assessed.

This review followed strict Cochrane recommendations and analyzed the strength of the evidence using the GRADE methodology [27]. Our systematic review is endorsed by the fact that we used a larger number of databases, recent data extraction, and the inclusion of different chronic musculoskeletal conditions, e.g., patellofemoral pain, low back pain, and the addition of hip strength results. This approach increased the accuracy of our estimates but had a potential limitation in increasing the heterogeneity in our meta-analysis. Thus, other high-quality studies may increase our certainty regarding the effectiveness of hip strengthening in chronic musculoskeletal conditions.

This systematic review has some strengths, including the high methodological quality [25]; however, there are some limitations: it was not possible to explore the sources of heterogeneity. In this context, future high-quality randomized controlled trials are needed, which may change our estimated effects. In addition, the search was not carried out in the most internationally recognized scientific document platforms (Web of Science and Scopus) or in the gray literature.

## 5. Conclusions

Our results show that hip strengthening is an autonomous active intervention to reduce the intensity of pain and/or the strength of patellofemoral pain. In addition, hip strengthening added to another active interventions can be beneficial in reducing pain intensity and disability in patients with low back pain. Finally, the evidence suggested a short-term isolated effect of hip strengthening on hip strength in patellofemoral pain and low back pain. Although uncertain, even with a tendency to recommend hip-strengthening exercise, the very low quality evidence indicates a greater need for more studies with high methodological quality. Surprisingly, the efficacy of hip strengthening at medium and long terms is still unknown in musculoskeletal conditions. This conclusion is likely to change with the publication of new high-quality randomized controlled trials.

## Figures and Tables

**Figure 1 diagnostics-12-02910-f001:**
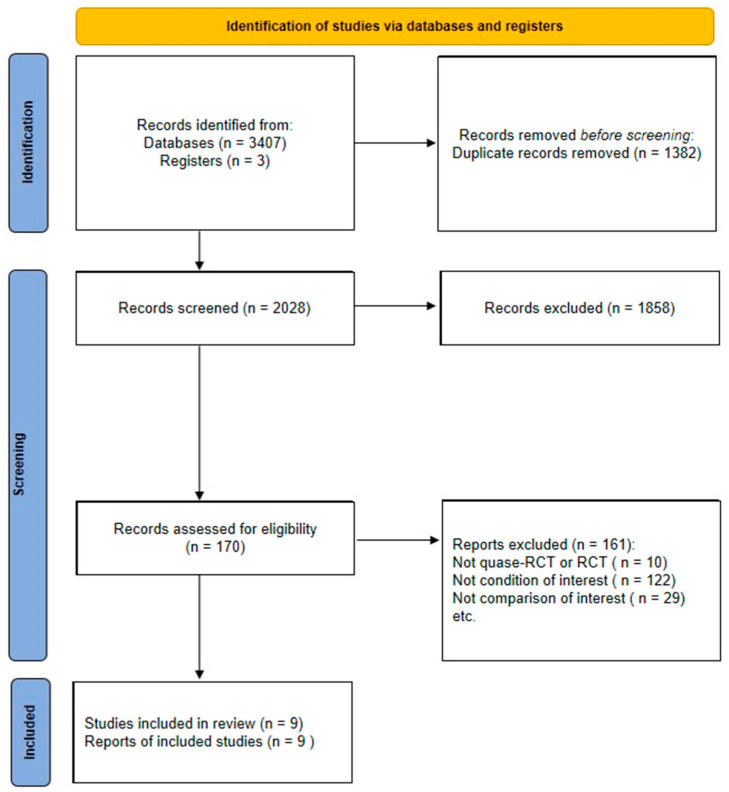
Flow of studies through the review. RCT: randomized clinical trial.

**Figure 2 diagnostics-12-02910-f002:**
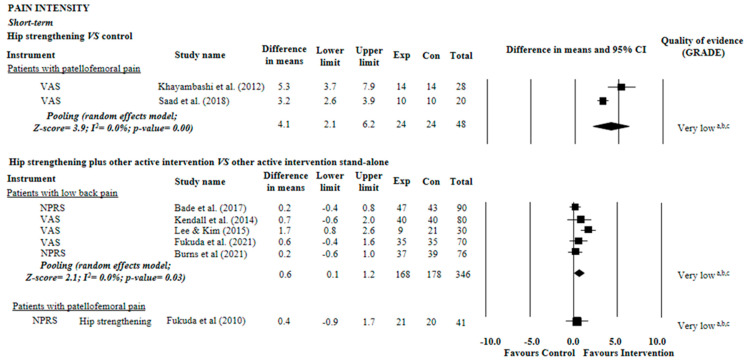
Summary of evidence of effect of hip strengthening on pain. Control: sham, placebo, no intervention, or waiting list. (a) Downgraded owing to imprecision: less than 400 participants included in the meta-analysis (sample of less than 200 was considered serious imprecision, and the evidence was downgraded by two levels); (b) Downgraded owing to inconsistency: I^2^ statistic was higher than 50% or pooling was not possible (poor overlap between the confidence intervals of the effects of the studies included in the meta-analysis was considered serious inconsistency, and the evidence was downgraded by two levels); (c) Downgraded owing to risk of bias: more than 25% of the participants in the meta-analysis were from trials with a high risk of bias (i.e., PEDro score < 6 of 10) [30,31,32,33,35,36,37,38].

**Figure 3 diagnostics-12-02910-f003:**
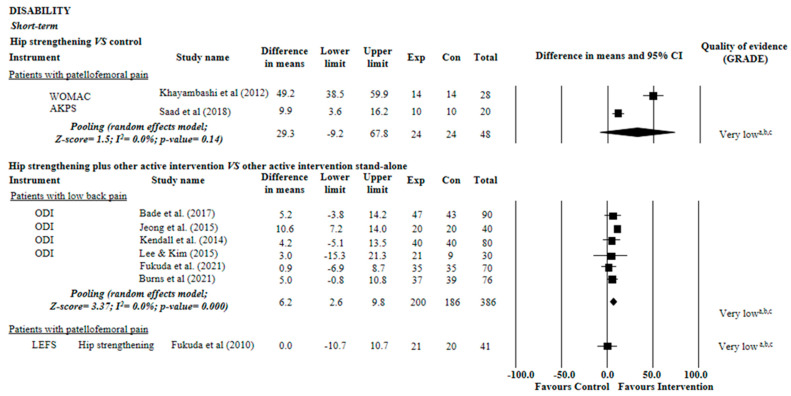
Summary of evidence of the effect of hip strengthening on disability. Control: sham, placebo, no intervention, or waiting list. (a) Downgraded owing to imprecision: less than 400 participants included in the meta-analysis (sample of less than 200 was considered serious imprecision, and the evidence was downgraded by two levels); (b) Downgraded owing to inconsistency: I^2^ statistic was higher than 50% or pooling was not possible (poor overlap between the confidence intervals of the effects of the studies included in the meta-analysis was considered serious inconsistency, and the evidence was downgraded by two levels); (c) Downgraded owing to risk of bias: more than 25% of the participants in the meta-analysis were from trials with a high risk of bias (i.e., PEDro score < 6 of 10) [30,31,32,33,34,35,36,37,38].

**Figure 4 diagnostics-12-02910-f004:**
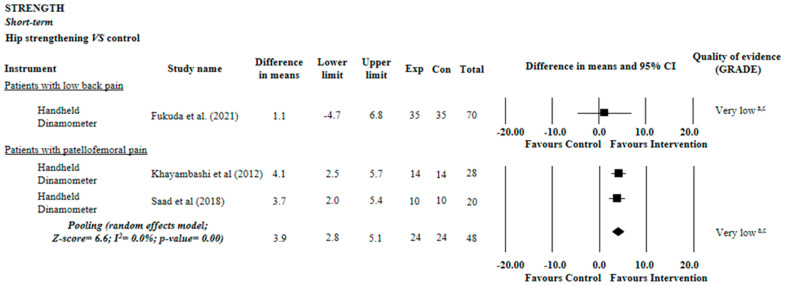
Summary of evidence of the effect of hip strengthening on strength. Control: sham, placebo, no intervention, or waiting list. (a) Downgraded owing to imprecision: less than 400 participants included in the meta-analysis (sample of less than 200 was considered serious imprecision, and the evidence was downgraded by two levels); (b) Downgraded owing to inconsistency: I^2^ statistic was higher than 50% or pooling was not possible (poor overlap between the confidence intervals of the effects of the studies included in the meta-analysis was considered serious inconsistency, and the evidence was downgraded in two levels); (c) Downgraded owing to risk of bias: more than 25% of the participants in the meta-analysis were from trials with a high risk of bias (i.e., PEDro score < 6 of 10) [31,32,37].

**Table 1 diagnostics-12-02910-t001:** Characteristics of the included trials (n = 9).

Study	Source	Participants	Intervention	Outcome Measures
Bade et al. (2017) [33]	Patients with low back pain Location: Germany	N = 90 Age 46.4 (SD 2.8) Gender M: 53 F: 37	Exp1 = lumbar strengthening (exercises for low back pain treatment) 2x/week, 50 min/session, over 2 weeks (n = 43, age: 48.1 (SD 2.4)) Exp2 = lumbar + hip strengthening (exercises for low back pain treatment associated with exercises to strengthen hip-stabilizing muscles) 2x/week, 50 min/session, over 2 weeks (n = 47, age: 44.8 (SD 2.3))	Pain intensity: NPRS (0–10) Disability: ODI (0–50) Follow-up: 2 weeks (short-term)
Burns et al. (2021) [38]	Patients with low back pain Location: multicenter	N = 76 Age 40.05 (SD 2.21) Gender M: 29 F: 47	Exp 1 = treatment of the lumbar spine only (LBO group) 2 to 3x/week, approximately 45 to 60 min/session (n = 39, age: 40.2 (SD 19.9)) Exp2 = lumbar spine and hip treatments (LBH group) 2 to 3x/week, approximately 45 to 60 min/session (n = 37, age: 39.9 (SD 18))	Pain intensity: NPRS (0–10) Disability: ODI (0–100) Follow-up: 7 weeks (short-term)
Fukuda et al. (2021) [37]	Patients with low back pain Location: Brazil	N = 70 Age 37.7 (SD 3.53) Gender S/N	Exp1 = a manual therapy and lumbar segmental stabilization group (MTLS) 2x/week, 30 min/session, over 5 weeks (n = 35, age: 35.2 (SD 12.5)) Exp2 = specific hip-strengthening exercises plus manual therapy and lumbar segmental stabilization group (MTLSHS) 2x/week, 45 min/session, over 5 weeks (n = 35, age: 40.2 (SD 12.4))	Pain intensity: VAS (0–10) Disability: Roland–Morris (0–24) Strength: Force dynamometer Follow-up: 5 weeks (short-term)
Fukuda et al. (2010) [30]	Patients with patellofemoral pain Location: Brazil	N = 66 Age 24.6 (SD 6.6) Gender M: 0 F: 66	Exp1 = knee strengthening (exercises to strengthen quadriceps) 3x/week, 50 min/session, over 4 weeks (n = 20, age: 25.0 (SD 6.0)) Exp2 = knee + hip strengthening (exercises to strengthen quadriceps and exercises to strengthen the hip abductor and lateral rotator muscles) 3x/week, 50 min/session, over 4 weeks (n = 21, age: 25.0 (SD 7.0)) Con = no intervention (n = 25, age 24.0 (SD 7.0))	Pain intensity: NPRS (0–10) Disability: LEFS (0–80) Follow-up: 4 weeks (short-term)
Jeong et al. (2015) [34]	Patients with low back pain Location: Korea	N = 40 Age 41.2 (SD 6.1) Gender M: 0 F: 40	Exp1 = lumbopelvic muscle + gluteus strengthening (exercises to strengthen gluteus) 3x/week, 50 min/session, over 6 weeks (n = 20, age: 41.2 (SD 5.5)) Exp2 = lumbopelvic muscle strengthening (exercises to strengthen lumbopelvic muscles) 3x/week, 50 min/session, over 6 weeks (n = 20, age: 41.2 (SD 6.7))	Disability: ODI (0–50) Follow-up: 6 weeks (short-term)
Kendall et al. (2014) [35]	Patients with low back pain Location: Brazil	N = 80 Age 37 (SD 35.5) Gender M: 0 F: 80	Exp1 = lumbopelvic muscle strengthening (focused on the performance of the motor skill of co-contracting the transversus abdominis, multifidus, and pelvic floor muscles), 6 weeks (n = 40, age: 33 (SD 33.4)) Exp2 = lumbopelvic muscle + hip strengthening (co-contracting the transversus abdominis, multifidus, and pelvic floor muscles associated with open and closed kinetic chain hip-strengthening exercises), 6 weeks (n = 40, age: 41 (SD 37.45))	Pain intensity: VAS (0–100) Disability: ODI (0–50) Strength: Force dynamometer Follow-up: 6 weeks (short-term)
Khayambashi et al. (2012) [31]	Patients with patellofemoral pain (PFP) Location: Iran	N = 28 Age 29.7 (SD 5.3) Gender M: 0 F: 28	Exp1 = hip strengthening (exercises to strengthen hip external rotator muscles) 3x/week, 30 min/session, over 8 weeks (n = 14, age: 28.9 (SD 5.8)) Con = no intervention (n = 14, age 30.5 (SD 4.8))	Pain intensity: VAS (0–10) Disability: WOMAC (0–100) Strength: Handheld isometric dynamometer Follow-up: 8 weeks (short-term)
Lee and Kim (2015) [36]	Patients with low back pain Location: Iran	N = 33 Age 60.46 (SD 14.4) Gender: S/N	Exp = hip strengthening + lumbar strengthening (exercises to strengthen hip, including flexion, extension, abduction, adduction, internal rotation, and external rotation) 3x/week, 20 min/session, over 6 weeks (n = 22, age:61.0 (SD 13.2)) Con = lumbar strengthening (exercises for lumbar stabilization) 3x/week, 20 min/session, over 6 weeks (n = 11, age:59.38 (SD 17.3))	Pain intensity: VAS (0–100) Disability: ODI (0–50) Follow-up: 3 weeks (short-term)
Saad et al. (2018) [32]	Patients with patellofemoral pain (PFP) Location: Brazil	N = 20 Age 22.85 (SD 1.1) Gender M: 0 F: 20	Exp2 = hip strengthening (exercises to strengthen hip-stabilizing muscles) 2x/week, 50 min/session, over 8 weeks (n = 10, age: 22.5 (SD 1.08)) Con = no intervention (n = 10, age 23.2 (SD 1.03))	Pain intensity: VAS (0–10) Disability: AKPS (0–100) Strength: Handheld isometric dynamometer Follow-up: 8 weeks (short-term)

**Table 2 diagnostics-12-02910-t002:** PEDro Scale Scores for individual trials * (n = 9).

Study	1	2	3	4	5	6	7	8	9	10	Total
Bade et al. (2017) [33]	Yes	No	Yes	No	No	No	Yes	No	Yes	Yes	5
Burns et al. (2021) [38]	Yes	Yes	No	No	No	No	No	Yes	Yes	Yes	5
Fukuda et al. (2021) [37]	Yes	Yes	Yes	No	No	No	Yes	Yes	Yes	Yes	7
Fukuda et al. (2010) [30]	Yes	Yes	Yes	No	Yes	Yes	Yes	No	Yes	Yes	8
Jeong et al. (2015) [34]	Yes	No	No	No	No	No	Yes	No	Yes	Yes	4
Kendall et al. (2014) [35]	Yes	Yes	Yes	No	Yes	Yes	Yes	Yes	Yes	Yes	9
Khayambashi et al. (2012) [31]	Yes	Yes	No	No	No	No	Yes	No	Yes	Yes	5
Lee and Kim (2015) [36]	Yes	No	Yes	No	No	No	Yes	No	Yes	Yes	5
Saad et al. (2018) [32]	Yes	Yes	Yes	No	Yes	Yes	Yes	Yes	Yes	Yes	9
Total, n (%)	9 (100)	6 (66.6)	6 (66.6)	9 (100)	3 (33.3)	3 (33.3)	8 (88.8)	4 (44.4)	9 (100)	9 (100)	

Abbreviation: PEDro, Physiotherapy Evidence Database (scores range from 0 to 10). 1, random allocation; 2, concealed allocation; 3, baseline comparability; 4, blinding of subjects; 5, blinding of therapists; 6, blinding of assessors; 7, more than 85% follow-up; 8, intention-to-treat analysis; 9, reporting of between-group statistical comparisons; 10, reporting of point measures and measures of variability. * Criterion 1 was not added to the total score, which is out of 10. Median, 5; interquartile range, 4; range, 4 to 9.

## Data Availability

The data presented in this study are available on request from the corresponding author.

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
