# Peer review of "Efficacy of Hip Strengthening on Pain Intensity, Disability, and Strength in Musculoskeletal Conditions of the Trunk and Lower Limbs: A Systematic Review with Meta-Analysis and Grade Recommendations"

_diagnostics, 2022, doi:10.3390/diagnostics12122910_

Round 1
Reviewer 1 Report (Previous Reviewer 1)
The Abstract is an honest summary of a well designed and implemented review report.
I have not major concerns, but for the general readers additional references are needed in the introduction and discussion:
A few examples follow below:
Latessa I, Ricciardi C, Jacob D, Jónsson H Jr, Gambacorta M, Improta G, Gargiulo P. Health technology assessment through Six Sigma Methodology to assess cemented and uncemented protheses in total hip arthroplasty. Eur J Transl Myol. 2021 Mar 9. doi: 10.4081/ejtm.2021.9651. Epub ahead of print.
Norouzi A, Behrouzibakhsh F, Kamali A, Yazdi B, Ghaffari B. Short-term complications of anesthetic technique used in hip fracture surgery in elderly people. Eur J Transl Myol. 2018 Aug 9;28(3):7355. doi: 10.4081/ejtm.2018.7355. eCollection 2018 Jul 10.
Magnússon B, Pétursson Þ, Edmunds K, Magnúsdóttir G, Halldórsson G, Jónsson HD Jr, Gargiulo P. Improving Planning and Post-Operative Assessment for Total Hip Arthroplasty. Eur J Transl Myol. 2015 Mar 11;25(2):4913. doi: 10.4081/ejtm.2015.4913. eCollection 2015 Mar 11.
.
Author Response
Answers to Reviewers
Manuscript number: diagnostics-1959629
REVIEWERS’ COMMENTS:
REVIEWER #1
The Abstract is an honest summary of a well designed and implemented review report.
I have not major concerns, but for the general readers additional references are needed in the introduction and discussion:
A few examples follow below:
Latessa I, Ricciardi C, Jacob D, Jónsson H Jr, Gambacorta M, Improta G, Gargiulo P. Health technology assessment through Six Sigma Methodology to assess cemented and uncemented protheses in total hip arthroplasty. Eur J Transl Myol. 2021 Mar 9. doi: 10.4081/ejtm.2021.9651. Epub ahead of print.
Norouzi A, Behrouzibakhsh F, Kamali A, Yazdi B, Ghaffari B. Short-term complications of anesthetic technique used in hip fracture surgery in elderly people. Eur J Transl Myol. 2018 Aug 9;28(3):7355. doi: 10.4081/ejtm.2018.7355. eCollection 2018 Jul 10.
Magnússon B, Pétursson Þ, Edmunds K, Magnúsdóttir G, Halldórsson G, Jónsson HD Jr, Gargiulo P. Improving Planning and Post-Operative Assessment for Total Hip Arthroplasty. Eur J Transl Myol. 2015 Mar 11;25(2):4913. doi: 10.4081/ejtm.2015.4913. eCollection 2015 Mar 11.
Answer: We thank you for the comment and included references as suggested

Reviewer 2 Report (Previous Reviewer 2)
This manuscript presents a synthesis of the knowledge on the efficacy of hip strengthening on pain intensity, disability and strength in musculoskeletal conditions of the trunk and lower limbs with the PRISMA methodology. The subject is of great relevance for the study field of musculoskeletal rehabilitation and physical therapy, however, the manuscript has several methodological weaknesses, so it cannot be published in its current state.
Major comments
1. Method:
• Authors need to include the most internationally recognized scientific document platforms, such as Web of Science and Scopus, in their search for studies. In this sense, it is not pertinent to contrast their findings with other systematic reviews that did include these platforms.
• It is necessary for the authors to present the studies found and selected in the PRISMA-2020 flowchart.
• It is necessary to include as a supplement a table with the references of the 163 full-text reviewed full-text articles that were excluded, specifying the reasons.
• It is necessary to include the PRISMA-2020 checklist as a supplement, to verify that the methodological criteria are met.
2. Discussion:
• The main problem and limitation of the study is the analysis of its findings.
• The authors acknowledge that the quality of the evidence is low or very low, however they suggest that the treatment is effective.
• Considering that the quality of the evidence of the studies is "low" or "very low", the performance of the meta-analysis is not justified. Therefore, the interpretation is inadequate.
• The discussion should be rewritten, since it is confusing and contradictory, the lack of scientific evidence should be highlighted and the treatment should not be recommended.
• The authors must recognize among the limitations of the systematic review that a search was not carried out in the most internationally recognized scientific document platforms (Web of Science and Scopus) and in the gray literature.
Minor comments
· In the page 12, line 302, the authors state "Only three of the nine included studies [20,29] assessed isolated hip strengthening and hip strength after the intervention." They need to include the missing reference.
Author Response
REVIEWER #2
This manuscript presents a synthesis of the knowledge on the efficacy of hip strengthening on pain intensity, disability and strength in musculoskeletal conditions of the trunk and lower limbs with the PRISMA methodology. The subject is of great relevance for the study field of musculoskeletal rehabilitation and physical therapy, however, the manuscript has several methodological weaknesses, so it cannot be published in its current state.
Major comments
- Method:
- Authors need to include the most internationally recognized scientific document platforms, such as Web of Science and Scopus, in their search for studies. In this sense, it is not pertinent to contrast their findings with other systematic reviews that did include these platforms.
Answer: We thank the Reviewer’s comment and accommodated suggestions. We believe they have improved the manuscript quality. We followed the Cochrane recommendations (https://training.cochrane.org/handbook/current/chapter-04) and conducted optimized search strategies in 8 databases: MEDLINE, COCHRANE (Central Register of Controlled Trials – CENTRAL and Database of Systematic Reviews), EMBASE, AMED, CINAHL, SPORTDISCUS and PEDRO. In addition, we hand searched reference lists to identify potential trials not identified in the search strategy. Therefore, our search strategy followed rigorous methods as acknowledged by the AMSTER II (https://www.bmj.com/content/358/bmj.j4008).
- It is necessary for the authors to present the studies found and selected in the PRISMA-2020 flowchart.
Answer: We thank the Reviewer’s comment and accommodated suggestion.
- It is necessary to include as a supplement a table with the references of the 163 full-text reviewed full-text articles that were excluded, specifying the reasons.
Answer: We accommodated suggestion in Appendix 2 on the Addenda.
- It is necessary to include the PRISMA-2020 checklist as a supplement, to verify that the methodological criteria are met.
Answer: We accommodated suggestion as a supplement in Appendix 1 on the Addenda.
- Discussion:
- The main problem and limitation of the study is the analysis of its findings.
- The authors acknowledge that the quality of the evidence is low or very low, however they suggest that the treatment is effective.
- Considering that the quality of the evidence of the studies is "low" or "very low", the performance of the meta-analysis is not justified. Therefore, the interpretation is inadequate.
Answer: We clarify that meta-analysis is conducted to quantify effect estimates, when possible, using random-effects model (Der-Simonian and Laird method). When it is not possible, data from individual trials are reported. It followed the strict Cochrane recommendations and AMSTAR II for systematic reviews of efficacy of interventions, and is according to our protocol registered prospectively. We also clarify that the GRADE methodology used in our review does not estimate effects, but the certainty of the current evidence; whether future high quality trials are likely to change the estimates. We added reference for the GRADE approach, and assessed it following recommendations and registered protocol. In our review, we found promessing effects; however, current quality of the evidence (low or very low) suggested that estimated effects may change in the future. It is clarifies in the Methods and Discussion sections.
- The discussion should be rewritten, since it is confusing and contradictory, the lack of scientific evidence should be highlighted and the treatment should not be recommended.
Answer: We revised the Discussion section as suggested and clarified our analysis in the comment above
- The authors must recognize among the limitations of the systematic review that a search was not carried out in the most internationally recognized scientific document platforms (Web of Science and Scopus) and in the gray literature.
Answer: We clarify that our optimized search strategy followed the Cochrane recommendations (https://training.cochrane.org/handbook/current/chapter-04) in 8 databases: MEDLINE, COCHRANE (Central Register of Controlled Trials – CENTRAL and Database of Systematic Reviews), EMBASE, AMED, CINAHL, SPORTDISCUS and PEDRO. In addition, we hand searched reference lists to identify potential trials not identified in the search strategy. It was according to our protocol registered prospectively. Therefore, we believe that our search strategy was not a limitation, as acknowledged by the AMSTER II (https://www.bmj.com/content/358/bmj.j4008).
Minor comments
- In the page 12, line 302, the authors state "Only three of the nine included studies [20,29] assessed isolated hip strengthening and hip strength after the intervention." They need to include the missing reference.
Answer: We changed as suggested (See page 12, line 292):
Only two of the nine included studies [31,32] assessed isolated hip strengthening (See page 12, line 2).

Reviewer 3 Report (New Reviewer)
Thank you for the opportunity to review your article.
You tried to perform a metaanalyse by 9 articles on RCTs on "hip strengthening". Your finding is - as I conclude - that there is no good evidence for anything. This is not surprising. A small number of RCTs on a real heterogenic topic (plevic dysfunction and low back pain, hip pain, knee pain...). Furthermore "hip strengthening" is not a conclusive intervention at all, because numerous interventions can be done on numerous pelvic muscles.
Author Response
REVIEWER #3
Thank you for the opportunity to review your article.
You tried to perform a metaanalyse by 9 articles on RCTs on "hip strengthening". Your finding is - as I conclude - that there is no good evidence for anything. This is not surprising. A small number of RCTs on a real heterogenic topic (plevic dysfunction and low back pain, hip pain, knee pain...). Furthermore "hip strengthening" is not a conclusive intervention at all, because numerous interventions can be done on numerous pelvic muscles.
Answer: As noted in the background, previous reviews were limited in both scope and quality, which in some cases rendered them misleading. Our systematic review followed current methodological guidance from Cochrane and the GRADE working group to update and synthesize the evidence on efficacy. By reporting comparable effect estimates and a rating of the certainty of evidence, this review provides patients and clinicians with reliable information to contribute to their decision-making processes. In addition to the improvement in methodological quality of the review itself, we have updated the literature. We feel that this improvement in methodological standard for systematic reviews is vital in this field where competing therapies, many of extraordinarily little value, are commonly used in clinical practice.

Reviewer 4 Report (New Reviewer)
The paper is a systematic review and meta-analysis of hip strengthening exercises on pain and disability of musculoskeletal conditions of the trunk and lower limbs. The authors highlighted that very low quality evidence suggested short-term effect of exercise on hip pain intensity and hip strength. The paper is interesting, however I have the some comments for the authors.
Major:
- Figure 3 and Page 9, lines 226-228. The authors have pooled the WOMAC score and AKPS. Although both measure patients disability I do not think that the authors could pool the results of the two different tests to meta-analyze them.
Minors:
- Please correct the reference errors (page 2, line 47; page 3, lines 126-127 and 136-137; page 4, line 152; page 12, line 319).
- Please remove lines 66-74 in page 2.
- Page 2, line 86. Appendix 1 is not available for review.
- Figure 1. It is not clear how from 1382 records the authors screened 2028 items. Please correct the error.
- Page 5, line 176; Table 1; and figure 2. Please define NPRS, otherwise use NRS as stated in page 4, line 100.
- Figure 3 and Page 9, lines 226-228. The authors have pooled the WOMAC score and AKPS. Although both measure patients disability I do not think that the authors could pool the results of the two different tests to meta-analyze them.
Author Response
REVIEWER #4
The paper is a systematic review and meta-analysis of hip strengthening exercises on pain and disability of musculoskeletal conditions of the trunk and lower limbs. The authors highlighted that very low quality evidence suggested short-term effect of exercise on hip pain intensity and hip strength. The paper is interesting, however I have the some comments for the authors.
Major:
- Figure 3 and Page 9, lines 226-228. The authors have pooled the WOMAC score and AKPS. Although both measure patients disability I do not think that the authors could pool the results of the two different tests to meta-analyze them.
Answer: We thank the Reviewer’s comment. We clarify that we pooled trials with enough clinical and methodological homogeneity (including our outcomes of interest assessed with different valid instruments), following the Cochrane recommendations and our prospectively registered protocol. The Anterior Knee Pain Scale (AKPS) and the Western Ontario and McMaster Universities (WOMAC) are valid instruments for assessment of disability in PFP.
Minors:
- Please correct the reference errors (page 2, line 47; page 3, lines 126-127 and 136-137; page 4, line 152; page 12, line 319).
Answer: We accommodated the suggestion.
- Please remove lines 66-74 in page 2.
Answer: We accommodated the suggestion.
- Page 2, line 86. Appendix 1 is not available for review.
Answer: We submitted the Appendix 1.
- Figure 1. It is not clear how from 1382 records the authors screened 2028 items. Please correct the error.
Answer: We clarified the flowchart (Figure 1). From 3,410 identified records, 1,382 duplicates were removed and 2,028 titles and abstracts were screened.
- Page 5, line 176; Table 1; and figure 2. Please define NPRS, otherwise use NRS as stated in page 4, line 100.
Answer: We standardized Numerical Perception Rating Scale as NPRS throughout the manuscript.
- Figure 3 and Page 9, lines 226-228. The authors have pooled the WOMAC score and AKPS. Although both measure patients’ disability, I do not think that the authors could pool the results of the two different tests to meta-analyze them.
Answer: As clarified above, we pooled trials with enough clinical and methodological homogeneity (including our outcomes of interest assessed with different valid instruments), following the Cochrane recommendations and our prospectively registered protocol. The Anterior Knee Pain Scale (AKPS) and the Western Ontario and McMaster Universities (WOMAC) are valid instruments for assessment of disability in PFP

Reviewer 5 Report (New Reviewer)
Silva_Hip strenght_diagnostics_2022
I commend the authors on the completion of this manuscript. Overall it is well written and on an important topic. I have a few concerns highlighted below.
General comments
Please, revision of the references. Multiple reference sources not found in the text.
Introduction
Lines 66-74. Please remove.
4. Discussion
Lines 276-278: “Thus, our data analysis and inclusion criteria may have led to the lower between-study heterogeneity observed in our estimates compared with the previous reviews that downgraded evidence due to inconsistency.” Please, syntax review. In the actual manuscript, inconsistency, thought maybe in a lower degree, has also downgraded evidence.
Lines 285-287: “Of note, even in a very short-term strength-training 285 program, we expected in hip strength gains due to increased muscle activation and firing 286 frequency, in addition to the synchronization of motor units and reduced co-activation of 287 antagonist muscles during exercise.” Please, syntax review.
Author Response
REVIEWER #5
Silva_Hip strenght_diagnostics_2022
I commend the authors on the completion of this manuscript. Overall it is well written and on an important topic. I have a few concerns highlighted below.
General comments
Please, revision of the references. Multiple reference sources not found in the text.
Answer: We thank the Reviewer’s comment and have revised the references.
Introduction
Lines 66-74. Please remove.
Answer: We accommodated the Reviewer’s comment.
- Discussion
Lines 276-278: “Thus, our data analysis and inclusion criteria may have led to the lower between-study heterogeneity observed in our estimates compared with the previous reviews that downgraded evidence due to inconsistency.” Please, syntax review. In the actual manuscript, inconsistency, thought maybe in a lower degree, has also downgraded evidence.
Answer: We accommodated the Reviewer’s comment:
“Thus, our data analysis and inclusion criteria may have led to the lower between-study heterogeneity observed in our estimates compared with the previous reviews”.
Lines 285-287: “Of note, even in a very short-term strength-training program, we expected in hip strength gains due to increased muscle activation and firing frequency, in addition to the synchronization of motor units and reduced co-activation of antagonist muscles during exercise.” Please, syntax review.
Answer: We accommodated the Reviewer’s comment:
“Notably, even in a very short-term strength-training program, we expected increases in hip strength because of neuromuscular changes”.

Round 2
Reviewer 2 Report (Previous Reviewer 2)
The authors responded to comments and significantly improved the manuscript.
Minor comments
I reiterate my recommendation.
• The authors must recognize among the limitations of the systematic review that a search was not carried out in the most internationally recognized scientific document platforms (Web of Science and Scopus) and in the gray literature.
Author Response
Answers to Reviewers
Manuscript number: diagnostics-1959629
REVIEWER 2
Minor comments
I reiterate my recommendation.
- The authors must recognize among the limitations of the systematic review that a search was not carried out in the most internationally recognized scientific document platforms (Web of Science and Scopus) and in the gray literature.
Answer: We thank the Reviewer’s comment and accommodated suggestion as follows (highligthed in green):
This systematic review has some strengths, including the high methodological quality [25]; however, there are some limitations: it was not possible to explore sources of heterogeneity. In this context, future high quality randomized controlled trials are needed, which may change our estimated effects. In addition, the search was not carried out in the most internationally recognized scientific document platforms (Web of Science and Scopus) and in the gray literature.

Reviewer 3 Report (New Reviewer)
Thank you for your revision. You made the best out of poor conditions.
Author Response
Answers to Reviewers
Manuscript number: diagnostics-1959629
REVIEWER 3
Thank you for your revision. You made the best out of poor conditions.
Answer: We thank the Reviewer’s comment.

This manuscript is a resubmission of an earlier submission. The following is a list of the peer review reports and author responses from that submission.
Round 1
Reviewer 1 Report
For the general readers some additional REFERENCES MUST BE ADDED.
Examples are:
Ciliberti FK, Cesarelli G, Guerrini L, Gunnarsson AE, Forni R, Aubonnet R, Recenti M, Jacob D, Jónsson H Jr, Cangiano V, Islind AS, Gambacorta M, Gargiulo P. The role of bone mineral density and cartilage volume to predict knee cartilage degeneration. Eur J Transl Myol. 2022 Jun 28;32(2):10678. doi: 10.4081/ejtm.2022.10678.
1. Recenti M, Ricciardi C, Edmunds K, Jacob D, Gambacorta M, Gargiulo P. Testing soft tissue radiodensity parameters interplay with age and self-reported physical activity. Eur J Transl Myol. 2021 Jul 12. doi: 10.4081/ejtm.2021.9929. Epub ahead of print.
1. Latessa I, Ricciardi C, Jacob D, Jónsson H Jr, Gambacorta M, Improta G, Gargiulo P. Health technology assessment through Six Sigma Methodology to assess cemented and uncemented protheses in total hip arthroplasty. Eur J Transl Myol. 2021 Mar 9. doi: 10.4081/ejtm.2021.9651. Epub ahead of print.
CT- and MRI-Based 3D Reconstruction of Knee Joint to Assess Cartilage and Bone.
Ciliberti FK, Guerrini L, Gunnarsson AE, Recenti M, Jacob D, Cangiano V, Tesfahunegn YA, Islind AS, Tortorella F, Tsirilaki M, Jónsson H Jr, Gargiulo P, Aubonnet R. CT- and MRI-Based 3D Reconstruction of Knee Joint to Assess Cartilage and Bone. Diagnostics (Basel). 2022 Jan 22;12(2):279. doi: 10.3390/diagnostics12020279. PMID: 35204370; PMCID: PMC8870751.
Reviewer 2 Report
I conseder that this article is outside the scope of the journal, since it focuses on the specific efficacy of a treatment and not on diagnostic tests. In addition, this manuscript has several methodological limitations in the systematic review.
Major comments
1. The authors did not use the PRISMA 2020 criteria criteria, they did not include the updated flowchart (the version included is that of PRISMA 2009).
2. The flowchart included is confusing, please review the numbers "Records screened n=2,028, Records excluded n=1,382, and decide to review 172 full text" what happened to the other 646 studies?
3. An exhaustive review of the gray literature was lacking. The authors should at least explore the clinical trials registries (ClinicalTrials.gov, and WHO’s International Clinical Trials Registry Platform) and one grey literature database (ProQuest Dissertation & Theses) to compliance the currency, completeness, and quality of this systematic review.
Reviewer 3 Report
The methodology of this study is appropriate, however, the structure of the content is not approroite, such as not clearly written of study hypothesis, limitation and conclusion. As the authors admitted the only very low quality stuides were remained after the screening, so the meta analysis should not be done.